# The First Nationwide Surveillance of Severe Fever with Thrombocytopenia Syndrome in Ruminants and Wildlife in Taiwan

**DOI:** 10.3390/v15020441

**Published:** 2023-02-05

**Authors:** Chih-Ying Kuan, Tsai-Lu Lin, Shan-Chia Ou, Shih-Te Chuang, Jacky Peng-Wen Chan, Ken Maeda, Tetsuya Mizutani, Ming-Pin Wu, Fan Lee, Fang-Tse Chan, Chao-Chin Chang, Rui-Ling Liang, Sue-Fung Yang, Tsung-Ching Liu, Wu-Chun Tu, Hau-You Tzeng, Chia-Jung Lee, Chuen-Fu Lin, Hsu-Hsun Lee, Jhih-Hua Wu, Hsiao-Chien Lo, Kuan-Chieh Tseng, Wei-Li Hsu, Chi-Chung Chou

**Affiliations:** 1Graduate Institute of Microbiology and Public Health, National Chung Hsing University, Taichung 40227, Taiwan; 2New Taipei City Government Animal Protection and Health Inspection Office, New Taipei City 110022, Taiwan; 3Department of Veterinary Medicine, College of Veterinary Medicine, National Chung Hsing University, Taichung 40227, Taiwan; 4National Institute of Infectious Disease, Shinjuku City, Tokyo 162-0052, Japan; 5Center for Infectious Diseases Epidemiology and Prevention Research, Tokyo University of Agriculture and Technology, Fuchu City, Tokyo 183-8509, Japan; 6Tainan City Animal Health Inspection and Protection Office, Tainan 730064, Taiwan; 7Animal Health Research Institute, Council of Agriculture, Executive Yuan, New Taipei City 251203, Taiwan; 8Endemic Species Research Institute, Jiji, Nantou 552005, Taiwan; 9Department of Animal Science and Biotechnology, Tunghai University, Taichung 407224, Taiwan; 10Department of Entomology, National Chung Hsing University, Taichung 402204, Taiwan; 11Livestock Research Institute, Council of Agriculture, Executive Yuan, Hsinhua, Tainan 712009, Taiwan; 12Department of Veterinary Medicine, National Pingtung University of Science and Technology, Neipu, Pingtung 912301, Taiwan; 13Hengchun Branch, Institute of Livestock Research, Council of Agriculture, Executive Yuan, Hengchun, Pingtung 946008, Taiwan; 1420 Cent Equine Practice, Kaohsiung 807020, Taiwan; 15Chu Lu Ranch, Beinan, Taitung 954401, Taiwan

**Keywords:** severe fever with thrombocytopenia syndrome, ruminant, wildlife, Taiwan

## Abstract

Since the first discovery of severe fever with thrombocytopenia syndrome virus (SFTSV) in China in 2009, SFTSV has rapidly spread through other Asian countries, including Japan, Korea, Vietnam and Pakistan, in chronological order. Taiwan reported its first discovery of SFTSV in sheep and humans in 2020. However, the prevalence of SFTSV in domestic and wildlife animals and the geographic distribution of the virus within the island remain unknown. A total of 1324 animal samples, including 803 domestic ruminants, 521 wildlife animals and 47 tick pools, were collected from March 2021 to December 2022 from 12 counties and one terrestrial island. The viral RNA was detected by a one-step real-time reverse transcription polymerase chain reaction (RT-PCR). Overall, 29.9% (240/803) of ruminants showed positive SFTSV RNA. Sheep had the highest viral RNA prevalence of 60% (30/50), followed by beef cattle at 28.4% (44/155), goats at 28.3% (47/166), and dairy cows at 27.5% (119/432). The bovine as a total of dairy cow and beef cattle was 27.8% (163/587). The viral RNA prevalence in ticks (predominantly *Rhipicephalus microplus*) was similar to those of ruminants at 27.7% (13/47), but wild animals exhibited a much lower prevalence at 1.3% (7/521). Geographically the distribution of positivity was quite even, being 33%, 29.1%, 27.5% and 37.5% for northern, central, southern and eastern Taiwan, respectively. Statistically, the positive rate of beef cattle in the central region (55.6%) and dairy cattle in the eastern region (40.6%) were significantly higher than the other regions; and the prevalence in Autumn (September–November) was significantly higher than in the other seasons (*p* < 0.001). The nationwide study herein revealed for the first time the wide distribution and high prevalence of SFTSV in both domestic animals and ticks in Taiwan. Considering the high mortality rate in humans, surveillance of other animal species, particularly those in close contact with humans, and instigation of protective measures for farmers, veterinarians, and especially older populations visiting or living near farms or rural areas should be prioritized.

## 1. Introduction

Severe fever with thrombocytopenia syndrome (SFTS) is an emerging tick-borne infectious disease caused by the etiologic agent, SFTS virus (SFTSV). SFTSV was first classified as the genus *Phlebovirus* and has been officially designated as *Dabie bandavirus* belonging to the genus *bandavirus* in 2020 by the International Committee on Taxonomy of Viruses 2020 [1]. However, colloquially it was referred to more often as the well-recognized and commonly used designation of SFTSV owing to its clinical features [2].

Since the first discovery of SFTSV in Henan province, China, in 2009 [3], SFTSV rapidly spread through other Asian countries, including Japan and Korea in 2013 [4,5], Vietnam in 2018 [6,7], Taiwan in 2019 [8], and Pakistan in 2020 [9]. In China, there are 7721 confirmed human SFTS cases, with 810 deaths from 2012 to 2018 [10]. Moreover, there were 763 SFTS cumulative cases in Japan by July 2022 [11], and a total of 1089 confirmed cases with 214 deaths were reported in South Korea [12]. In addition to Asia, two patients with SFTS-like illness were reported in Missouri, United States in 2012. The pathogen was named Hartland virus and confirmed to be phylogenetically related to SFTSV [13]. Although some studies hypothesized that migratory birds carrying ticks are responsible for the transboundary spread of the virus [14], the route of SFTS transmission remains inconclusive.

SFTSV caught attention owing to its high fatality rate in humans. Studies showed that the mortality rate could be as high as 30% [15]. Clinical manifestations of SFTS in humans include high fever, vomiting, gastrointestinal disorders, weight loss, thrombocytopenia, leukopenia, and multiple organ failure. Since no specific therapeutic treatment for SFTS is satisfactorily effective, early diagnosis plays an important role in patient survival and overall clinic outcomes. Given that the clinical symptoms of SFTS are mostly indistinguishable from many infectious diseases, history of outdoor activities (particularly tick bites) and laboratory confirmations are critical for SFTS diagnosis [16].

Tick infestation is the main route of SFTS transmission in nature. In endemic areas, viral RNA has been detected in many species of ticks, including *Haemaphysalis flava*, *Rhipicephalus microplus*, *Amblyomma testudinarium*, *Dermacentor nuttalli*, and *Hyalomma asiaticum* [2]. In order to maintain SFTSV survival, animals had been proposed as the amplifying hosts to sustain a tick-to-animal cycle in nature. SFTSV RNA and SFTSV-specific antibodies have been detected in several animal species, including sheep, cattle, chicken, ducks, pigs, dogs and cats, with sero- and RNA prevalence ranging from 2.6% to 97.9% [17,18,19]. In herbivores, Niu et al. found 60.5–69.5% seroprevalence and 3.8–4.2% RNA prevalence in Shandong Province, China [18]. Huang et al. also found that in Henan Province, China, the seroprevalence and RNA prevalence were as high as 69.7–97.9% and 15.8–18.8%, respectively [19]. This literature suggested that herbivores usually have relatively higher SFTSV prevalence than other domestic animals.

As a vector-borne disease, accumulated evidence indicated that human-to-human transmission or animal-to-human transmission could also occur through direct contact [20]. Yamanaka et al. demonstrated that two veterinarians acquired SFTSV by close contact with the affected cat rather than via cat scratch or tick bite [21]. Kobayashi et al. conducted an epidemiology study of SFTS cases from 2012 to 2017 in Japan; almost half of the patients had a history of companion animal contacts [22], which emphasizes that personnel with animal-related occupations, including veterinarians, ranch workers, and farmers could be under higher biosafety risk [23]. It is known that animals infected with SFTSV exhibit moderate disease symptoms or remain asymptomatic [18,24], rendering authorities less vigilant to the occurrence of SFTS in animals. As a zoonotic disease with a high fatality rate in human beings, surveillance of the SFTSV in suspected animals sharing the same habitat as humans is an important means to mitigate the potential threats to humans.

In Taiwan, SFTS in animals and humans was not identified until 2019–2020 [8,25], despite the fact that the infection status of SFTSV in Japan, South Korea and China has been well documented. In the first SFTS report in Taiwan, the SFTSV RNA was detected in 29% and 4.8% of the sheep and bovine specimens in one farm [8]. In the same year, the first human fatal case was also reported [25] and officially considered a locally acquired SFTS case. Of note, only one free-ranging farm was under surveillance in the first report, and the geographical distribution of SFTSV and the infectious status of domestic animals in Taiwan remains completely unknown. Hence, the current study implemented nationwide surveillance of SFTSV investigating the prevalence of viral RNA in the ruminants from both free-ranging and intensive housing farms. The results should contribute to the knowledge gap in the geographical distribution of SFTSV along the east Asia coastline countries, as well as the infection status in both animals (domestic and wildlife) and vectors in relation to the transmission route and zoonotic cycle of SFTSV.

## 2. Materials and Methods

### 2.1. Sample Collection

In total, 1324 animal samples were collected for this study. The samples included 803 domestic animal serums, of which 587, 166, and 50 specimens belonged to cattle, goats, and sheep, respectively. The other 521 were wildlife samples, including 258 wild birds and 263 mammalian animals, that were referred to the Animal Health Research Institute (225 birds, 237 mammals) or rescued by the Endemic Species Research Institute (33 birds, 20 mammals), Council of Agriculture, Taiwan. Additionally, 6 wild boar samples were provided by licensed hunters. Moreover, 47 tick pools collected from cattle (41 pools) and wild boars (6 pools) were also analyzed. The experimental protocol and sample collection of animals were approved by the Institutional Animal Care and Committee of National Chung Hsing University (IACUC number: 109-060R).

### 2.2. Detection of Viral RNA by One-Step Real-Time Reverse Transcription Polymerase Chain Reaction (RT-PCR)

The total RNA of the animal serum samples was isolated by Maxwell RSC simplyRNA Tissue kit (Promega, Madison, Wisconsin, USA), while tick samples were initially homogenized by TissueLyser II (QIAGEN, Venlo, The Netherlands), followed by RNA extraction using Trizol^®^ reagent (Invitrogen, Carlsbad, CA, USA). Subsequently, 0.5 μg of total RNA was reverse-transcribed with iTaq™ Universal SYBR^®^ Green One-Step kit (Bio-Rad, Hercules, CA, USA) with primers (SFTSV-SF: 5′-ACCTCTTWGACCCTGAGTTWGACA-3′, SFTSV-SR: 5′-CTRAAGGAGACAGGTGGAGATGA-3′) as reported from Centers for Disease Control [25]. The amplification conditions were: incubation at 42 °C for 30 min, followed by 95 °C, 30 s, and further amplification: 30 s at 95 °C, 60 s at 60 °C for 40 cycles and final melting curve program from 68 °C to 95 °C. Hence, the positive case was defined by the Cq value < 38 and was further validated by the melting curve analysis and by the gel electrophoresis.

### 2.3. Amplification of Partial S Segment for Automated Sequencing

For SFTSV-positive samples, the sequence of the partial S segment was amplified by nested PCR. Briefly, the cDNA was generated by superscript IV (Invitrogen, Carlsbad, CA, USA), and that served as a template for nested PCR. The first run of amplification used outer set primers (1F: 5′-CATCATTGTCTTTGCCCTGA-3′, 1R: 5′-AGAAGACAGAGTTCACAGCA-3′), and 2 μL of the amplicon from the first run PCR then served as a template for the second run amplification using the inner set primer (2F: 5′-AAY AAG ATC GTC AAG GCA TCA-3′, R: 5′-TAG TCT TGG TGA AGG CAT CTT-3′). The PCR was conducted with the following condition: 5 min at 95 °C, followed by 40 cycles of 30 s at 95 °C, 45 s at 60 °C and 1 min at 72 °C. The PCR product was isolated, and the identity of the amplicon was revealed by automated sequencing (Mission Biotechnology, Taipei, Taiwan).

### 2.4. Phylogenetic Analysis

The seven representative partial sequences of the S segment (Accession numbers: OQ362401-OQ362407) were aligned by the CLUSTAL W software of the MegAlign program. Phylogenetic analysis of the partial S segment of represented samples was conducted by the Maximum Likelihood method, and Kimura 2-parameter model, with 1000 bootstrap replicates.

### 2.5. Statistical Analysis

The association of viral RNA positivity with risk variables including animal species, geographical areas, the prevalence in each season, and herd management, was analyzed by using a chi-squared test in IBM SPSS Statistics 20.

## 3. Results

### 3.1. Prevalence of SFTSV RNA on Ruminant Samples

The number of ruminant samples and their collection site/county was illustrated in Figure 1. A total of 803 serum samples from farms located in Northern, Central, Western, Southern and Eastern Taiwan, as well as a terrestrial island (Kinmen), was demonstrated. Domestic animals in this study consisted of 50 sheep, 166 goats, and 587 cattle (432 dairy cows and 155 beef cattle).

Overall, 240 of the 803 serum samples showed SFTSV RNA positive, a prevalence of 29.9% in total ruminants. Sheep have the highest viral RNA prevalence of 60% (30/50), followed by beef cattle at 28.4% (44/155), goats at 28.3% (47/166), and dairy cows at 27.5% (119/432), in descending order. In caprine, the prevalence in sheep is significantly higher than in goats (*p* < 0.001). The bovine as a total of dairy cow and beef cattle was 27.8% (163/587) (Table 1). However, statistically, the overall prevalence between beef and dairy cattle showed no difference. Among the farms analyzed, only one herd (goat) was absent of SFTSV RNA.

SFTS could be found in almost all regions of the nation. Geographically the distribution of positivity was quite even. Positive animals were identified in 33% (16/48) of the northern, 29.1% (83/285) of the central, 27.5% (88/320) of the southern, and 37.5% (45/120) of the eastern regions (Table 2), with the eastern region having the highest SFTSV prevalence among the 5 main island regions. In beef cattle, RNA positive rate in the central region (55.6%) was significantly higher than the other regions (*p* < 0.001). On the other hand, the rate of dairy cows in the eastern region (40.6%) was significantly higher than farms in the other regions (*p* < 0.001). Of note, the positive rate in Kinmen island (the nearest region to China) was 26.7% (8/30).

As SFTS is a tick-borne disease, seasonal dynamics were further analyzed. Of the 803 ruminant samples, SFTSV RNA prevalence of samples collected in Spring (March–May), Summer (June–August), Autumn (September–November), and Winter (December–February) was 26% (106/408), 27.7% (52/188), 47.2% (60/127) and 27.5% (22/80), respectively (Figure 2). Statistically, the prevalence of samples collected in Autumn is significantly higher than those in other seasons (*p* < 0.001).

### 3.2. Prevalence of SFTSV RNA on Wildlife Samples

The number of wildlife samples and their collection city/county is illustrated in Figure 3. The 521 wildlife animal samples (serum or tissue) included 263 mammals and 258 birds collected from all over the nation. Wild mammals included 6 wild boars and 257 other mammals (Formosan macaque, Formosan ferret-badger, Formosan muntjac, leopard cat, masked palm civet, pangolin, squirrel, crab-eating mongoose, otter, weasel, and yellow-throated marten). Overall, only 7 samples from 2 wild boars, 4 mammals, and 1 bird were detected positive for SFTSV RNA. The SFTSV prevalence in wild boars, other mammals, and birds were 33% (2/6), 1.6% (4/257), and 0.4% (1/258), respectively. It is worth noting that most of the wildlife samples (462 animals) were postmortem tissues, and only 59 were blood samples drawn from the rescued animals. When the quality of the sample was taken into account and excluded postmortem tissues, the positive rate in serum samples of wildlife was 11.9% (7/59).

### 3.3. Prevalence of SFTSV RNA on Ticks

Ticks collected from participating animals were grouped into 47 pools, of which 41 were from cattle and 6 from wild boars. Ticks on the cattle were all identified as *Rhipicephalus microplus*, while ticks on the wild boars were identified as *Haemaphysalis hystricis* and *Amblyomma testudinarium*. Overall, SFTSV RNA was detected from 13 pools (all from cattle), and the prevalence rate was 27.7%. Of note, all the positive samples (i.e., *Rhipicephalus microplus*) were obtained from beef cattle, and the prevalence of ticks collected from cattle was 31.7% (13/41).

### 3.4. Sequence Analysis of Partial S Segment of SFTSV

The seven partial S sequences identified from sheep, goats, cows and ticks from the current study (labeled with a triangle in Figure 4) were compared with human and animal strains isolated from other countries, including China, Japan and Korea. In general, the sequences among our local strains shared high similarity and were alternately placed in the same clade (Figure 4), suggesting a close phylogenetic relationship.

## 4. Discussion

SFTS is a tick-borne zoonotic disease with a wide range of animal hosts and can cause high mortality in humans [10,26,27]. However, not until the first reported human SFTS fatal case in 2019 [25] was Taiwan considered as an endemic area. In order to understand the potential risk of SFTSV in Taiwan and to update the distribution status of SFTSV in Asia, it is desirable to investigate the epidemiology of SFTSV in domestic and wild animals in Taiwan. This was the first nationwide surveillance of SFTSV infection in ruminants and wildlife animals, the major natural reservoirs of SFTSV in Taiwan. The results implied a high prevalence of SFTSV infection in Taiwan.

The epidemiology of SFTSV varies among different animal species and countries. Domestic ruminants such as sheep, cattle, and goats showed relatively high prevalence as compared with other animal species, and they are regarded as the amplifying hosts [18,28]. The SFTSV has been extensively studied in China, and according to previous studies in Shandong and Henan provinces, the viral RNA prevalence in sheep was from 3.8–15.79% [18,19]. Investigation herein revealed that sheep had the highest RNA prevalence (60%), which was significantly higher (*p* < 0.05) than the goats and other animal species tested in this study. It is worth mentioning that the sheep herd surveyed in our local farm (in Nantou county) was managed for touristic purposes, where human-to-animal and animal-to-animal contacts are recurring. Since this is also the farm we reported the first discovery of SFTSV in Taiwan from [8], the persistent occurrence of the virus is established. In addition, beef cattle in the same farm had the highest positive rate (55.6%) among all cattle herds analyzed. Therefore, it is possible that during a long period of cohabitation, sustainable circulation of SFTSV between the vector and the host animals in the pasture happened and led to increased prevalence, but supporting data from a more completed seasonal survey is needed for confirmation. Moreover, SFTSV RNA prevalence in another free-range farm (in Taitung county, Eastern Taiwan) was 40.6%, the highest prevalence among the herds of dairy cows. These results indicate that grazing in grasslands, which increases exposure to ticks carrying the virus, could be a risk factor associated with SFTSV. Noticeably, the average RNA prevalence of cattle was 27.8%, which was slightly above that reported in the Shandong and Henan provinces of China (4.16–18.75%, respectively) [18,19]. Similarly, 28.3% of goats in the current study carried SFTSV RNA, which was significantly higher than the previous reports (i.e., 2–2.4%) in the South Korea [29,30]. By backtracking the sampled animals, most of the domestic animals in our study were kept and fed in free-stall housing; the high density and intensive housing could facilitate the spread of the virus. Moreover, an attempt was made to enrich viral nucleic acid content in serum samples following one previous report [31]; cellular DNA and RNA were digested with a cocktail of DNase and RNase enzymes, while viral nucleotide capsulated by viral envelope could be protected from nuclease digestion. Hence, cellular DNA/RNA can be eliminated before the viral RNA is extracted, and the detection of a trace quantity of RNA could be enhanced with this additional procedure.

As a tick-borne disease, *Haemaphysalis longicornis* is suspected as the major species responsible for SFTS transmission based on reports from other countries [32]. However, consistent with our previous report [8], *R. microplus* was the predominant species of ticks infesting cattle in Taiwan. Of note, among the 3 species of ticks identified in this study, only *R. microplus* was detected positive for SFTSV, and the 6 pools of ticks from the wild boars (*Haemaphysalis hystricis* and *Amblyomma testudinarium*) all tested negative despite that *Amblyomma testudinarium* has been reported to be a possible vector [2]. In fact, the ticks that carry SFTSV might be different among different countries depending on the major tick species that inhabit the region, which is associated with the climate condition optimal for their living. On the same note, it is reported that the epidemic season of SFTSV in ruminants is from March through November, in which ticks are more active. In the current study, the peak incidence of SFTSV was from September to November, which is slightly later than the reported months from other counties, for instance, June to July in China [3]. We suspect that the discrepancy could also be attributed to the difference in overall climate and the species of tick in the regions. Nevertheless, the tick species detected and the overall seasonal variations were comparable to other Asian regions with SFTSV epidemics.

In regard to the spread of SFTSV, wild animals usually play a more important role than domestic animals due to their wider range of activity, carrying vectors with the virus from their original habitats to other ecosystems [33]. In the current study, viral nucleotide seroprevalence was relatively lower in wild animals than in domestic animals. As previously reported, SFTSV was detected in wild animal species, including wild boar, masked palm civet, macaque, badger, deer, elk, hedgehog, and raccoon dog [28,34,35]. In the present study, in addition to wild boar and masked palm civet, the SFTS viral RNA was also detected in the pangolin and oriental honey-buzzard. The three mammals above are common wild animal species in Taiwan and inhabit plains to mid-elevation altitude [36,37,38], covering cultivated fields and orchards that overlap the areas of human activities. Such living patterns may contribute not only to a higher chance of virus spread through their movements, but also to human infection by possible contact. Besides mammals, the only avian species detected positive for SFTSV RNA was the oriental honey-buzzard. The migratory birds carrying ticks harboring SFTSV have been highly suspected to be responsible for the transboundary spread of SFTSV [14,39]. In Taiwan, the oriental honey-buzzards may stay for breeding and then migrate to different countries as the season transitions. Their long-distance migratory routes cross through China, South Korea, Japan, the Philippines, and Malaysia [40,41], which coincides with the SFTS endemic areas along the east Asian coastline. However, further studies are needed to elucidate the role of oriental honey-buzzards as a natural reservoir spreading SFTS overseas.

The results in the present study provide ample information that is useful to better understand the current situation of SFTSV in Taiwan. The collection of samples from 12 cities/counties represented a good geographical coverage such that the positive results could indicate a wide distribution of SFTSV in both domestic and wild animal species in Taiwan with confidence. Moreover, RNA prevalence in ticks is also high at 27.7%, making it feasible to sustain the SFTSV life cycle in nature. However, in view of the geographical latitude of Taiwan, whether or not the wide SFTSV distribution could be attributed to ticks alone warrants further study. Nevertheless, our discovery is unprecedented in Taiwan, and the higher-than-expected prevalence all around Taiwan should prompt the implementation of biosecurity measures. Monitoring for SFTS in susceptible animals and occupational personnel, including caretakers, farmers, veterinarians, and especially the older populations visiting or living near the farm or rural area, is eminent.

## 5. Conclusions

This was the first nationwide surveillance of SFTSV infection in animals in Taiwan. A high prevalence of viral RNA in both domestic farm animals and wildlife animals was discovered. Since SFTSV RNA was readily detected in samples of almost all regions, it suggested that this zoonotic infectious disease has already been widely distributed in Taiwan. Based on these findings, animal health takers and people working in animal-related occupations should be made aware of the potential risk associated with exposure to SFTSV-infected animals. Proper attention and protective measures should be instigated by governmental officials and policymakers from both the veterinary and human medicine sides in Taiwan. Moreover, surveillance of wider host ranges and those with closer contact with humans, especially companion animals, is warranted; the investigation of potential new vector species spreading the disease in warmer regions of the continent is strongly recommended.

## Figures and Tables

**Figure 1 viruses-15-00441-f001:**
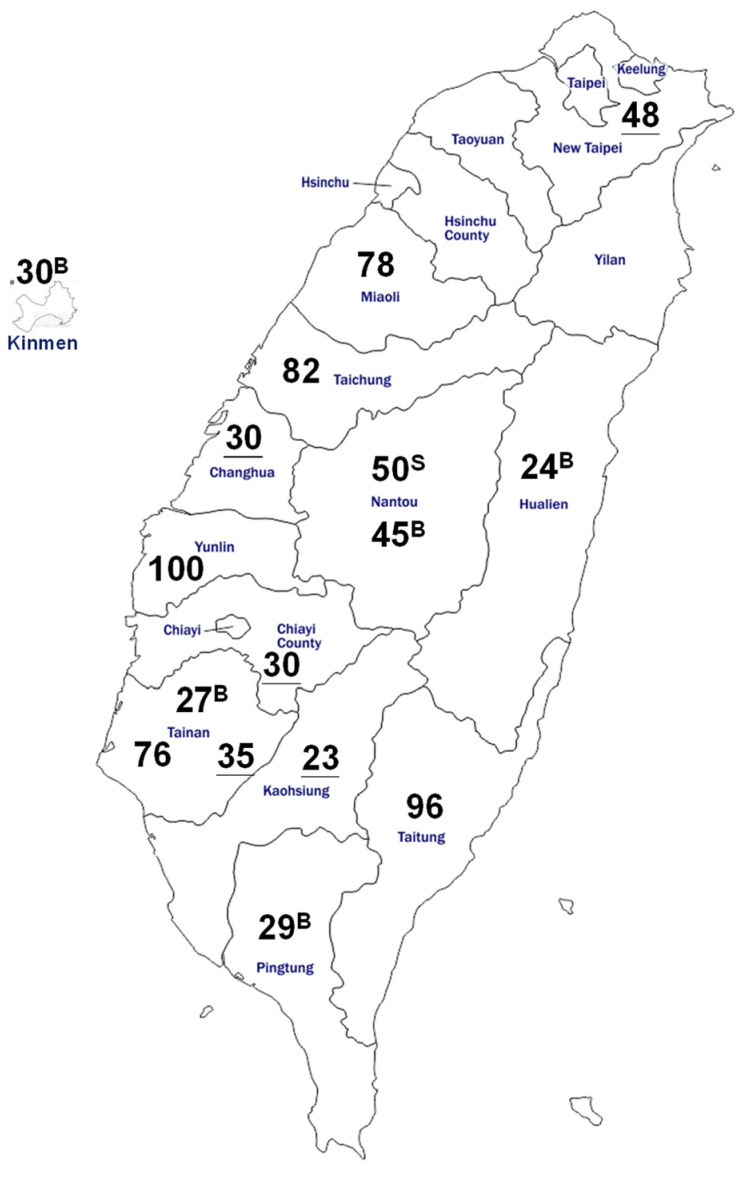
Map of Taiwan showing the location and number of ruminants enrolled in the study for SFTSV viral RNA detection. In total, 803 serum samples were collected from dairy cows, beef cattle, goats, and sheep in 15 cities in Taiwan. The numbers with specific symbols (superscript letters B, S, or underlined) indicate beef cattle, sheep, or goats, respectively, while the number of dairy cows is shown without any symbol.

**Figure 2 viruses-15-00441-f002:**
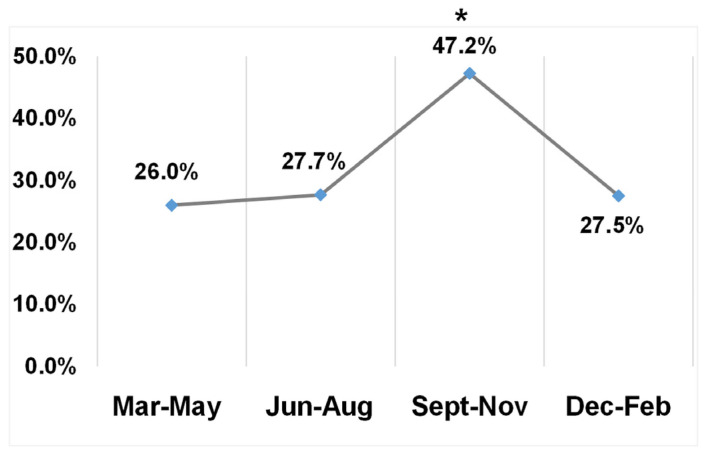
The prevalence of SFTSV RNA in ruminant samples in each season. The four seasons were defined as March–May, June–August, September–November, and December–February for Spring, Summer, August, and Winter, respectively. The prevalence of each season was noted by the numbers. Significantly higher prevalence was noted in samples collected in the Autumn (* indicated *p* < 0.001).

**Figure 3 viruses-15-00441-f003:**
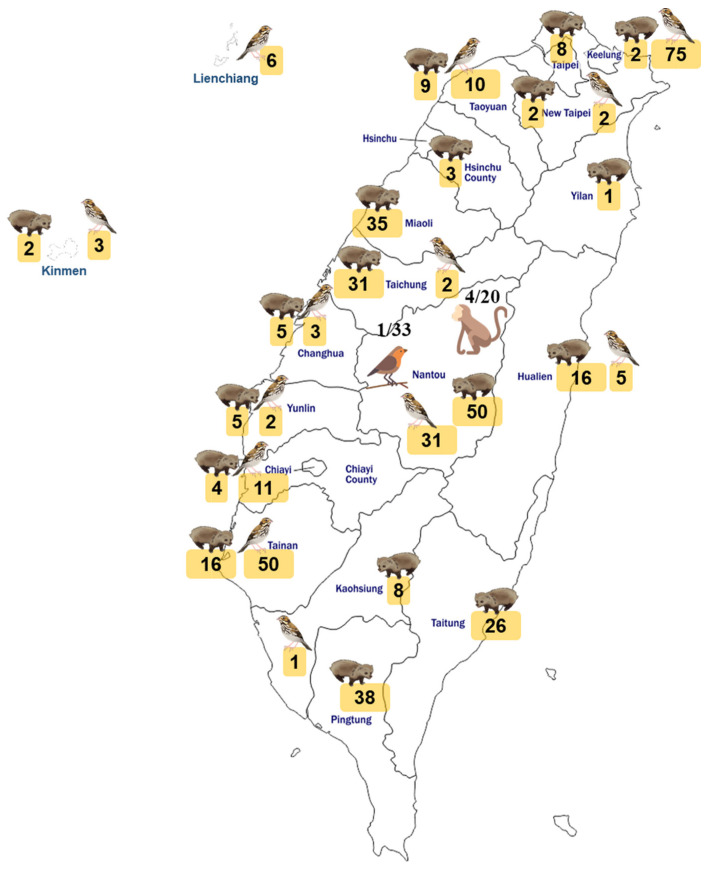
Map of Taiwan showing the location and number of wildlife enrolled in the study for SFTSV viral RNA detection. In total, 521 serum or tissue samples were collected from 263 mammals, and 258 birds were collected from all over the nation in 18 cities of Taiwan.

**Figure 4 viruses-15-00441-f004:**
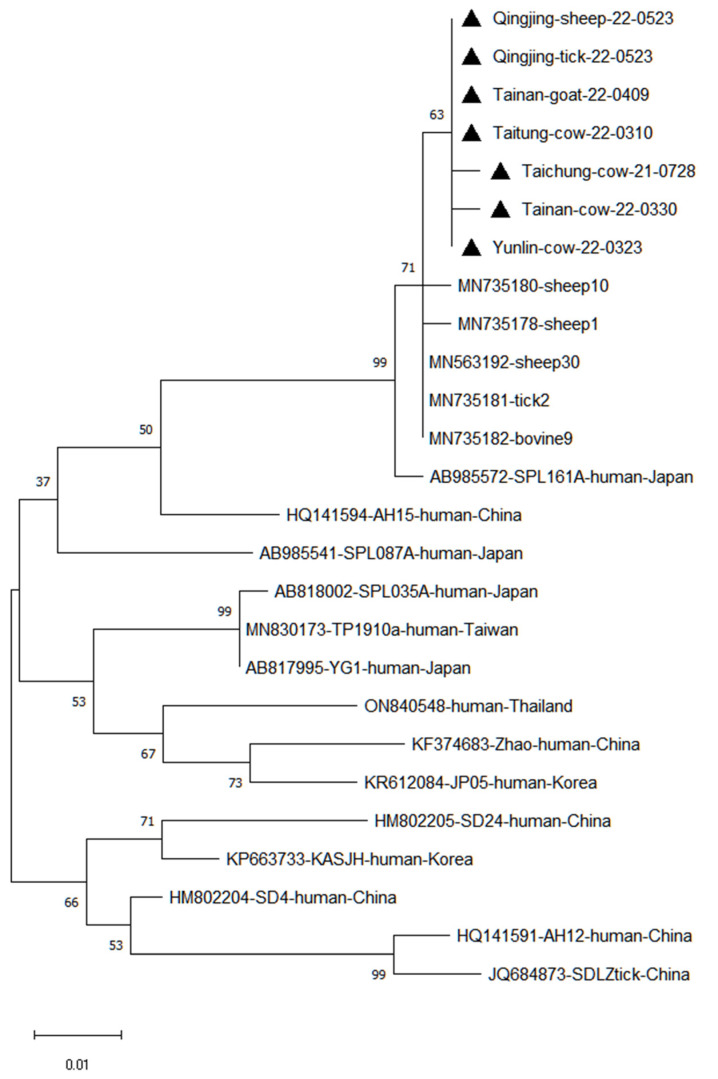
Phylogenetic analysis of the partial S segment of SFTSV identified in Taiwan. In total, sequences of 7 positive samples (indicated as a black triangle) with higher RNA load were amplified and further analyzed. Other representative viral strains were presented with their accession numbers and also the host and country of isolation. The evolutionary history was inferred using the maximum-likelihood method, based on the Kimura 2-parameter model (1000 bootstrap replicates). The percentage of trees in which associated taxa clustered is shown next to the branches. The scale bar indicates nucleotide substitutions per position.

**Table 1 viruses-15-00441-t001:** Prevalence of SFTS viral RNA on animals.

Samples	Ruminants				Wildlife			Vector	
	Caprine		Cattle		Mammals	Boar	Bird	Ticks	
	Sheep	Goat	Dairy	Beef				On Cattle	On Boar
Viral RNA (number/positive rate)	50 (60%)	166 (28.3%)	432 (27.5%)	155 (23.2%)	257 (1.6%)	6 (33%)	258 (0.4%)	41 (31.7%)	6 (0%)

**Table 2 viruses-15-00441-t002:** Prevalence of SFTS viral RNA on ruminants in each city/county.

	City/County	Species	Sample Size	RNA Prevalence
Northern	New Taipei	goat	48	33% (16/48)
Central	Miaoli	dairy cow	78	7.7% (6/78)
	Taichung	dairy cow	82	19.5% (16/82)
	Changhua	goat	30	20% (6/30)
	Nantou	sheep	50	60% (30/50)
		beef cattle	45	55.6% (25/45)
Southern	Yunlin	dairy cow	100	38% (38/100)
	Chiayi	goat	30	46.7% (14/30)
	Tainan	goat	35	31.4% (11/35)
	Tainan	beef cattle	27	11.1% (3/27)
	Tainan	dairy cow	76	26.3% (20/76)
	Kaohsiung	goat	23	0% (0/23)
	Pingtung	beef cattle	29	6.9% (2/29)
Eastern	Hualien	beef cattle	24	25% (6/24)
	Taitung	dairy cow	96	40.6% (39/96)
Remoted island	Kinmen	beef cattle	30	26.7% (8/30)
Total			803	28.9% (240/803)

## Data Availability

Data sharing is not applicable.

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
