# Peer review of "The First Nationwide Surveillance of Severe Fever with Thrombocytopenia Syndrome in Ruminants and Wildlife in Taiwan"

_viruses, 2023, doi:10.3390/v15020441_

Round 1

Reviewer 1 Report

MS. Review.

Journal: Viruses

Manuscript ID; Viruses-2163614

Title; The First Nationwide Surveillance of Severe Fever with Thrombocytopenia Syndrome in Ruminants and Wildlife in Taiwan

Authors; Chih-Ying Kuan et al., 

In this manuscript, the authors demonstrated the high prevalence of SFTSV in both domestic farm animals and wild animals. This is an important report to raise awareness of the risk of an animal-borne SFTS epidemic in humans in Taiwan. On the other hand, the result and discussion part must be improved with more detail analysis and more explanations. Furthermore, English sentences are inadequate in many areas, and there are a number of inconsistencies in notation and spelling errors. Therefore, the English text needs to be proofread. The paper provides interesting data, but it still needs a considerable revision to be acceptable for publication after major revision.

Major issues. 

1.     Information on sample collection 

As SFTS is a tick-borne disease, seasonal dynamics are also important information. However, information on the season when samples were collected, and analyses based on this information were not included. Authors should be added this information and the result of analysis. In addition, although comparisons of positive rates by animal species, area, etc. were made, no description of statistical analysis and the result of statistical analysis is found. Authors should be added the information. 

While authors reported the results of SFTSV prevalence in ticks surveyed, no detailed description of their numbers, species, timing of collected season and whether the tick was blood-sucking(engorged) or not. If the engorged ticks were positive, these should be discussed that their relationship with the host animal. 

2.     RNA analysis of SFTSV

The authors used SYBR green PCR system to detected SFTSV gene from samples. In such cases, it is possible to semi-quantify the viral copy number by referring to the host housekeeping gene. The authors discussed about the RNA detection level of this assay system in Discussion section (Line 230-233). If highly level of SFTSV gene detection in this report and the usefulness of the RNA assay used in this report were to be considered, the detection limit data of the PCR system should be stated.

Furthermore, in the manuscript of Material and Methods section, the authors described the sequences analysis targeting the S segment, but these results and analyses have not been carried out. 

3.     sustainable circulation of SFTSV

Authors discussed that “During a long period of cohabitation, sustainable circulation of SFSTV between the vector and the host animals in the pasture could lead to increased prevalence” in the Discussion section (Line216-217). However, the season in which the samples were collected was unknown and no consideration has been given to the duration of viremia level of SFTSV. If authors want to mention the possibility of “sustainable circulation” of SFTSV, references or the results of successive studies should be provided. 

Minor issues; 

Line39; Not in English text.

Line56,57; Viral scientific names should be italicized.

Line173; I could not know what this sentence is trying to convey.

Lien176; Not in English text.

Table 2; Animal species were not uniformly described. Initial letters should be lowercased. (Goat ==> goat)

Line178; site/country ==>city/country

Line196; SFSTV ==> SFTSV

Line216; SFSTV ==> SFTSV

Line269; the first evidence: Some group already reported the prevalence of SFTSV in human, animal, and ticks in Taiwan Furthermore, although the authors mentioned that the first evidence of SFTSV circulating in this study, no scientific data had presented to suggest that the SFTSV “circulating” is maintained. It was just “prevalence”.  

Author Response

Dear reviewer:
Please see the attachment.

Reviewer 2 Report

Major comments

1.     The high prevalence of only RNA detection in 1324 animal samples and 47 tick pools, while without another serologic detection or virus isolation, the reader might eager to know the sequences or genotypes are all the same or not among the positive sample of different animal species and different collecting sites. There should be some full gene sequences such as S segment in some different animal species or positive samples from different location to strengthen the results and to illustrate the possible related viral transmission, compare to  the similar SFTSV studies in China, Korea, and Japan.

2.     In conclusions section, the authors emphasize to investigate the potential new vector species spreading the SFTSV disease, however the tick species of 47 pools is not showed in this study.

Minor comments

1.     Line 39: “Overall, SFTSV prevalence in Ruminants was.”  à delete

2.     Line 57: The year 2020 is not consistent with ref [1].

3.     Line 134(F: “  àR: “

4.     Lne 135 : “ R: “  à ”F:”

5.     Line 159: “798”  à”803”

6.     Line 192: “521 serum samples ”  à ” 521 serum or tissues samples”

Line 226: “[28-30]”  à[29-30]”

Author Response

Dear reviewer,

Round 2

Reviewer 1 Report

The manuscript has been revised well.

I think this manuscript will be acceptable.